# Arabic translation and cultural adaptation of a training load and player monitoring in high-level football questionnaire: A cognitive interview study

Abdulmalek K. Bursais [ID] *

Department of Physical Education, College of Education, King Faisal University, Al-Ahsa, Saudi Arabia

* abursais@kfu.edu.sa

## Abstract

### Background

Understanding the current practice and the associated challenges in applying monitoring tools is essential to improving football performance in the Middle East, thus the purpose was to translate and culturally adapt a published questionnaire that assessed the practice and perception of High-Level football teams toward Training Load and Player Monitoring to be used in the Arabic context, aiming to contribute to the enhancement of football performance, player welfare, and training quality in the region.

### Method

A total of 15 Arabic-speaking coaches (mean age 42.6 ± 9.9 years; mean experience 10.9 ± 5.7 years; 53.3% football coaches and 46.7% strength & conditioning coaches) were conveniently selected to participate in this study. The current research followed a systematic cross-cultural adaptation process, which included forward translation, synthesis, back-translation, expert panel review, and pre-testing through cognitive interviewing. Three rounds of cognitive interviews were conducted with the 15 participants. Descriptive statistics, including means with standard deviations and frequencies with percentages, were reported for the participants' characteristics.

### Result

With some minor linguistic modifications to the questionnaire by the expert committee (i.e., adjustments such as *Sport Scientist* to *Sport Science Specialist*), the instrument was translated and culturally adapted into Arabic. All participants confirmed that the resulting Arabic versions of the training load and player monitoring in high-level football questionnaires were appropriate and fully understandable for Arabic speakers in conveying the intended meanings of the items in each.

**Data Availability Statement:** Data cannot be shared publicly because of [confidentiality and privacy]. Data are available from the Institutional Review Board (IRB), at King Faisal University, Al-

ahsa, KSA (contact via irb@KFU.edu.sa) for
researchers who meet the criteria for access to
confidential data.

**Funding:** This research was supported by the
Deanship of Scientific Research, Vice Presidency
for Graduate Studies and Scientific Research, King
Faisal University, Saudi Arabia [Project No.
GRANT6,157].

**Competing interests:** The authors have declared
that no competing interests exist.

## Conclusion

The training load and player monitoring in the high-level football questionnaire was successfully translated and culturally adapted into Arabic and are now ready for use in the Arabic context, offering an opportunity for comprehensive research and enabling tailored performance optimization strategies, which could ultimately lead to advancements in player development and welfare within Arabic-speaking football communities.

## Introduction

In a bid to reduce the occurrence of injuries and enhance performance, numerous elite football teams enlist the services of fitness and sport-science experts who diligently oversee the daily monitoring of training load (TL) [1]. TL is generally categorized into two components: external and internal training load. External training load refers to the physical exertion undertaken by the athlete, such as distance covered or number of sprints completed [2]. Internal training load, on the other hand, pertains to the physiological response induced by the training, including factors like heart rate and the athlete's perceived effort [2]. While there is evidence supporting the connection between training load and both performance and injury risk [3, 4], there remains a limited understanding of how these monitoring methods are practically applied within the realm of football.

Advancements in technology and analytical techniques have ushered in new opportunities within the applied sports environment. Nowadays, sports practitioners have the capability to monitor TL through the utilization of global navigation satellite systems and other microtechnology [5, 6]. In the past, elite and professional teams might have been hesitant to share their methods, but recent research indicates their increasing willingness to participate in and publish applied research [7, 8]. Offering an overview of the prevailing practices and viewpoints on monitoring will shed light on the difficulties experienced by practitioners and will inspire more relevant research within the industry [9, 10].

The Training Load and Player Monitoring in High-Level Football Questionnaire was developed by Akenhead & Nassis [11] offers probably the most comprehensive empirical analysis of information on the practices and practitioners' perceptions of monitoring in high-level professional football clubs. Their research revealed that factors such as human resources, low coach buy-in, and poor sensitivity of field measures have limited the effectiveness of training monitoring [11]. However, despite the advancements in sports technology, there is currently no consensus regarding which variables are the most valuable or how to effectively analyze the longitudinal data collected from a diverse roster of players [11]. Similarly, there exists limited knowledge regarding the current practices and the associated challenges in applying these methods within professional football, primarily due to their insufficient representation in the published literature.

While most of the research regards the training load and player monitoring was conducted in Europe, the United States, and Australasia [7, 11–13], yet nothing has been known about the practice of football teams in the Middle East (especially in Saudi Arabia). Recently, the Saudi Pro League (SPL) has received increased attention from all over the globe [14]. SPL officials aim to improve their League to be one of the world's leading football leagues [15]. Along with this growth, there is increasing interest in players' health and performance in the league [16]. Data about the efficacy, safety, and practice of player monitoring in the SPL is unknown; therefore, the availability of standardized measures in the Arabic context is warranted for elite

football. Given the absence of a valid and reliable Arabic questionnaire around this topic, the purpose of the current study was to undertake the translation and culture adaption of the Training Load and Player Monitoring in High-Level Football Questionnaire to be used in the Arabic context. This will help the development of representative samples and provide a set of standardized questionnaires for future use to compare data from different nations.

## Methods

A methodological cross-sectional designed to conduct the study. To assemble a participant group, a convenience sampling method was utilized, resulting in a total of 15 Arab coach participants, all of whom were aged over 32 years old from different Arab countries. Recruitment efforts spanned diverse locations within the Arab Region and were facilitated through the distribution of an online flyer, disseminated via platforms such as X (previously Twitter) and WhatsApp. Additionally, coaches working in Saudi Arabia during the period from July to September 2023 were approached for participation. Notably, individuals lacking fluency in the Arabic language were excluded from the study. This study was approved by the Institutional Review Board at King Faisal University, Alahsa, Saudi Arabia (IRP: ETHICS707), and participants provided written consent for their involvement.

### Instrument

This research details the comprehensive procedure involved in translating and culturally modifying the survey into Arabic for use with Arabic-speaking coaches. The original survey contained a total of 15 questions, encompassing 9 open-ended questions and 6 closed-ended ones. These questions pertained to various aspects such as the aims and methodologies of monitoring, analysis of data, and evaluations of the effectiveness of player monitoring. The instrument was made accessible on the International Journal of Sports Physiology and Performance's platform [11].

### Cross-cultural adaptation process

The present study adhered to the guidelines outlined for cross-cultural adaptation studies as recommended by [17], and was adapted in a previous study [18]. The process is illustrated in Fig 1.

### 1. Forward translation

Two proficient and unbiased bilingual translators proficient in both English and Arabic were engaged to conduct the initial translation of the English versions of the survey. Translator 1 possessed a familiarity with the subjects and themes addressed within the questionnaire. In contrast, Translator 2 was deliberately kept uninformed about the specific concepts being measured. This adherence to the guidelines ensured that Translator 2 ideally had no prior knowledge in the related field [17].

### 2. Synthesis of the translation

The author thoroughly reviewed the translated questionnaires generated in the initial phase, addressing and rectifying any discrepancies encountered. Following this, the author, along with the two translators, collaborated to generate a conclusive and mutually accepted version for each questionnaire, resulting in the creation of the final agreed-upon document known as the T12 questionnaire.

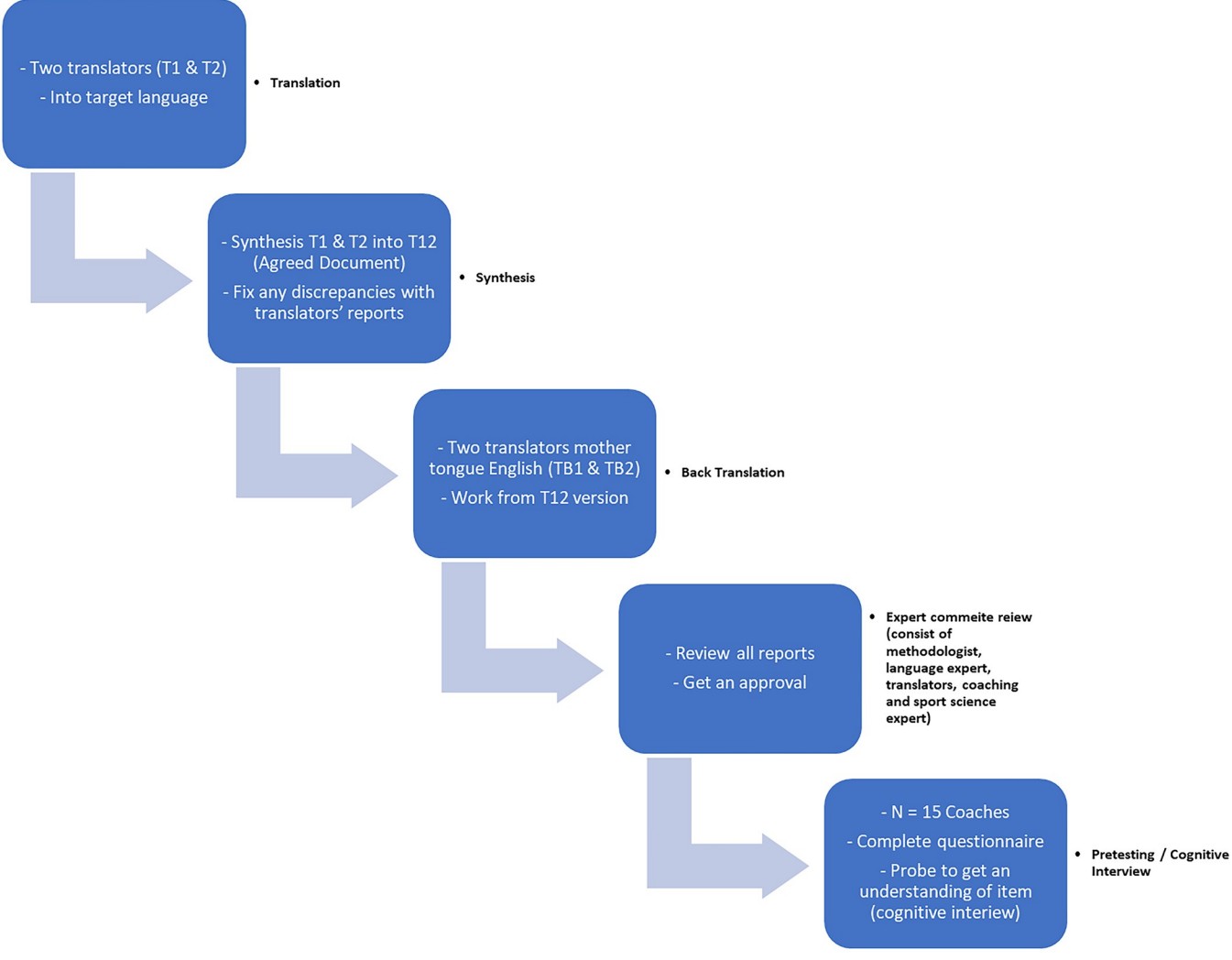

**Fig 1. Illustrates the translation and cross-cultural adaptation process as recommended by Beaton, et al. 2000.**

### 3. Back translation

The process of back translation for the T12 questionnaires was performed by two distinct bilingual independent translators. In this procedure, the translators retranslated the T12 questionnaire from the target language back into the original language (from Arabic to English), all while remaining unaware of the initial questionnaire or its subject matter. This back-translation phase serves the purpose of confirming that the questions are lucid, comprehensible, and accurately translated. The primary motivation behind this step was to prevent any potential information bias and uncover unforeseen nuances in the phrasing of items within the translated questionnaire, i.e., the T12 version.

### 4. Expert committee

The panel of experts consisted of five specialists well-versed in the domain of the instruments' knowledge and translation processes (forward and back translators). All members of the expert committee held doctoral degrees, possessed bilingual proficiency (Arabic-English), and

boasted substantial expertise in player monitoring, strength & conditioning, coaching, exercise physiology, sports science, and linguistics. The composition of this expert committee played a pivotal role in ensuring the achievement of satisfactory cross-cultural alignment within the research tools.

The complete set of original questionnaires, along with each translation (forward and backward), alongside corresponding written assessments composed by the author, were distributed to each expert committee member for thorough review. Their input was crucial in making informed judgments and validating the cross-cultural consistency between the original version and the final versions according to the guideline [17].

## 5. Pre-testing version

Fifteen coaches specializing in soccer or strength & conditioning, all of whom were native Arabic speakers (ages ranging from 30 to 65 years), participated in the completion of the questionnaires. Additionally, they engaged in cognitive interviews aimed at ascertaining their interpretations of the items within each questionnaire and the corresponding responses they selected, as per the provided definitions. The objective behind this pretesting phase of the novel questionnaire was to employ the pre-final version on participants within the intended context.

### Cognitive interviews

A cognitive interview serves as a qualitative technique designed specifically to examine the extent to which a survey question effectively accomplishes its intended purpose [19]. These cognitive interviews were conducted through face-to-face interactions or Zoom meetings, each held separately subsequent to a participant's completion of the final Arabic rendition of the translated questionnaires. Three rounds of cognitive interviews were conducted; five participants were involved in each round (see Table 1). These cognitive interviews aimed to validate that the respondents comprehended the questionnaire items and identify any potential need for rephrasing or restructuring. All interview queries were semi-structured, containing open-ended and closed-ended questions conducted entirely in Arabic. The role of the interviewer was assumed by the author in this context. The duration taken to address the questions was recorded. In each round, adjustments were made to the questions based on the feedback received from the participants. The subsequent content outlines the specific questions posed to participants during these cognitive interviews:

1. What do you think the question is about?

2. Is the question clear and understandable? If not, how can it be made clearer?

3. Do you have any questions about the items?

4. How could the wording be clearer?

### Statistical analysis

Descriptive data analysis was conducted for the study. Continuous variables were summarized using means along with their corresponding standard deviations (SD). For the cognitive interviews, the overall total score for each round was calculated as follows: The number of responses (each question in the cognitive interview) multiplied by 100, divided by the total number of answers. This provided the percentage of responses that were not flagged as having issues with their meaning. All data analyses were executed using IBM's SPSS version 27 (version 27.0; SPSS, Inc., Chicago, Illinois).

Table 1. Summary characteristics of the sample population (n = 15).

| Variables | R1 (n = 5) | R2 (n = 5) | R3 (n = 5) |
|---|---|---|---|
| | Mean + SD | | |
| Age (Years) | 37 ± 4.8 | 39.2 ± 9.4 | 51.6 ± 8.9 |
| Experience (Year) | 6 ± 8 | 10.6 ± 4.8 | 15.2 ± 6 |
| Coaching Type | | | |
| Football Coach | 2 | 3 | 3 |
| Strength & Conditioning Coach | 3 | 2 | 2 |
| Education | | | |
| High School | 2 | | |
| College Degree | 2 | 2 | 2 |
| Post-gradute Degree | 1 | 3 | 3 |
| Empolyment Status | | | |
| Unempolyed | 1 | | 1 |
| Part-time Coach | 2 | 2 | 3 |
| Full-time Coach | 2 | 3 | 1 |
| Nationality | | | |
| Saudi | 2 | 3 | 3 |
| Egyptian | 2 | 1 | 1 |
| Tunisian | 1 | 1 | 1 |

R = round, SD = standard deviation

Additional overall questions:

1. Are there any activities or examples that we omitted?

2. Did any of the questions make you feel uncomfortable?

## Results

### Cross-cultural adaptation

**Forward translation and synthesis of the translation.** Starting from the original survey, two separate skilled bilingual translators effectively converted the questionnaires into Arabic. Some slight differences in sentence structures arose due to the direct translation of certain English terms into Arabic. The disparities identified by the author and translators in the initial translation stage were effectively addressed. This was accomplished by clarifying the intended purpose of the questionnaire, leading to the creation of the T12 questionnaire version.

**Backward translation.** The T12 version of the questionnaire was translated from Arabic to English by two distinct independent bilingual translators. This process served as a means to ensure its validity; however, slight adjustments in synonym words and some sentence structures were identified within the questionnaire as a consequence of the translation of long sentences from English to Arabic, and subsequently resolved.

**Expert committee.** The Five experts reviewed the original questionnaires and the translated questionnaires together with the corresponding written reports provided by the author. The expert committee provided advice on necessary cultural modifications that align with the Arab region of the questionnaire items. For example, the Arabic language is not dominant in the leagues mentioned in the original version of the survey. Consequently, the experts proposed alternative leagues with Arabic dominance. Also, the expert panel suggested a few minor changes that could better align with the Arabic culture where, as an example, "*Sport Scientist*" was changed to "*Sport Science Specialist*". Table 2 details the minor changes provided by the expert committee, this alteration is acknowledged within the literature [18].

**Table 2. Minor changes provided by the expert committee of the training load and player monitoring in high-level football questionnaires Arabic version.**

| | Original item in the English version | Modified item in the Arabic version |
|---|---|---|
| 1 | Sport Scientist | Sport Science Specialist<br>أخصائي علوم رياضة |
| 2 | Strength & Conditioning Coach | Muscular Strength and Physical Preparation Coach<br>مدرب قوة عضلية وإعداد بدني |
| 3 | Rate of Perceived Exertion (RPE) | Borg's Perceived Exertion Scale<br>مقياس بورغ للإحساس بالجهد (RPE) |

**Table 3. Overall result of cognitive interview of the training load and player monitoring in high-level football questionnaires Arabic version.**

| Rounds | N | Participant Understanding of the Intended Meaning | The Content Was Clear for the Participant | The Wording Was Clear for the Participant |
|---|---|---|---|---|
| R1 | 5 | 100% | 100% | 100% |
| R2 | 5 | 100% | 100% | 100% |
| R3 | 5 | 100% | 100% | 100% |

## Cognitive interview

Fifteen football and strength & conditioning coaches who were native Arabic speakers aged between 32 and 55 years old participated in the interviews (see Table 1). The interviews consisted of three rounds of cognitive assessments, with five participants in each round responding to the questionnaires. The findings from these cognitive interviews conducted in Arabic are detailed in Table 3 (provided below). It was found that across all three rounds, all participants comprehended the intended meanings of the questionnaire items, achieving a 100% understanding rate. The collective outcomes underscore that every participant (100%) grasped the intended interpretations of the Arabic version's items throughout all rounds.

## Discussion

The availability of standardized measures capable of assessing the needs of Arabic-speaking coaches and players holds significant importance for enhancing sports practices [20]. Although most research instruments in sports have originated in English-speaking countries [21–24], various questionnaires have undergone translation and cultural adaptation from English to Arabic [19, 25–28]. Due to the absence of a suitable research instrument in Arabic language to evaluate the practice of training load and player monitoring in the Arab region [16], the current study aimed to translate and culturally adapt the Training Load and Player Monitoring in High-Level Football Questionnaire into Arabic to make the questionnaire items fully comprehensible to Arabic-speaking practitioners.

The present study has effectively accomplished the translation of the questionnaire from English into Arabic language. The questionnaire underwent a process of translation and cultural adaptation to guarantee equivalence between the original versions and the versions created in this study, as stipulated by guideline [17]. The Training Load and Player Monitoring in High-Level Football Questionnaire has been used in some major football leagues such as the English Premier League, La Liga, Serie A, and Bundesliga 1 [11], applying this questionnaire in Arab football leagues might help researchers, coaches, and stockholders generate evidence-based recommendations to improve the football in the region.

In this study, all participants comprehended the intended meanings of the questionnaire items achieving a 100% understanding rate in three rounds. The approach used in this

investigation is similar to that used by other researchers [18]. Alaqil et al. (2023) have translated and culture-adopted the Sedentary Behavior Questionnaire (SBQ), the Dietary Habits Questionnaire, and the Preclinical Mobility Limitation questionnaire for use in the Saudi Arabian context. After four rounds of cognitive interviews, participants confirmed that the resulting Arabic versions were suitable and entirely understandable for Arabic speakers, effectively giving the intended meanings of the items in each questionnaire [18]. The outcomes affirm that the new version has been successfully translated into Arabic and culturally adjusted to align with the Arabic context.

The key strengths of this study are the first to translate and validate a questionnaire that assesses the practice of player monitoring in football teams among Arabic-speaking practitioners. This study has also used different dialogues, backgrounds, and demographies during validation. The current study is limited by the fact that it did not assess the reliability of the Arabic questionnaire; however, future work is expected to cover this area. The availability of a valid Arabic-language version of the questionnaire for use with Arabic-speaking practitioners holds significant importance for research conducted in Middle Eastern countries, given the substantial prevalence of football interest in these regions [29, 30]. In summary, the outcomes affirm that the new version has been successfully translated into Arabic and culturally adjusted to align with the Arabic context.

## Supporting information

**S1 Questionnaire.**
(DOCX)

## Acknowledgments

The author would like to express his gratitude to all participants, translators, and the expert committee for their contributions to the cross-cultural process.

## Author Contributions

**Writing – original draft:** Abdulmalek K. Bursais.

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
