## [Decision Letter · Decision Letter 0]

22 Jan 2024

PONE-D-23-42649Arabic Translation and Cultural Adaptation of a Training Load And Player Monitoring in High-Level Football Questionnaire: A Cognitive Interview StudyPLOS ONE

Dear Dr. Bursais,

Thank you for submitting your manuscript to PLOS ONE. After careful consideration, we feel that it has merit but does not fully meet PLOS ONE’s publication criteria as it currently stands. Therefore, we invite you to submit a revised version of the manuscript that addresses the points raised during the review process.

We look forward to receiving your revised manuscript.

Kind regards,

Nour Amin Elsahoryi, pHD

Academic Editor

PLOS ONE

Journal Requirements:

"This research was supported by the Deanship of Scientific Research, Vice Presidency for Graduate Studies and Scientific Research, King Faisal University, Saudi Arabia [Project No. GRANT1445]."

Please state what role the funders took in the study.  If the funders had no role, please state: ""The funders had no role in study design, data collection and analysis, decision to publish, or preparation of the manuscript."" If this statement is not correct you must amend it as needed. 

"The author would like to express his gratitude to all participants, translators, and the

expert committee for their contributions to the cross-cultural process. The author would like to

thank the Deanship of Scientific Research, Vice Presidency for Graduate Studies and Scientific

Research, King Faisal University, Saudi Arabia for the financial support under Ambitious

Researcher track [Project No. GRANT1445]."

Funding information should not appear in the Acknowledgments section or other areas of your manuscript. We will only publish funding information present in the Funding Statement section of the online submission form. 

"This research was supported by the Deanship of Scientific Research, Vice Presidency for Graduate Studies and Scientific Research, King Faisal University, Saudi Arabia [Project No. GRANT1445]."

Reviewers' comments:

Reviewer's Responses to Questions

**Comments to the Author**

1. Is the manuscript technically sound, and do the data support the conclusions?

Reviewer #1: Partly

Reviewer #2: Yes

2. Has the statistical analysis been performed appropriately and rigorously? 

Reviewer #1: Yes

Reviewer #2: Yes

3. Have the authors made all data underlying the findings in their manuscript fully available?

Reviewer #1: Yes

Reviewer #2: Yes

4. Is the manuscript presented in an intelligible fashion and written in standard English?

Reviewer #1: No

Reviewer #2: Yes

5. Review Comments to the Author

Reviewer #1: Comments are provided in the attached pdf.

Summary of comments:

Abstract requires major changes to properly present the research work.

Introduction needs to be rewritten to have proper argument leading to the core of the study which is translation and culture adaptation of sports instruments. Also, there should be proper literature review with enough previous studies.

Methodology needs more elaboration.

Results needs more elaboration and more examples and numerical facts if possible.

Discussion is very poor. It should provide an elaborate interpretation of the results in relation to the research questions/objectives and previous research. Also, the limitations and recommendation offered are poor and require rewriting.

Reviewer #2: The manuscript is considered technically acceptable to the journal and supports the data and conclusions stated.

The manuscript is considered scientifically and technically sound. The experimentation process is sound and a modern idea.

Perform statistical analysis fairly appropriately.

The data provided is correct.

According to my humble experience, the manuscript is presented in a clear manner and written in, I believe, sound language.

6. PLOS authors have the option to publish the peer review history of their article (what does this mean?). If published, this will include your full peer review and any attached files.

Reviewer #1: **Yes: **Ahmed Alaa

Reviewer #2: **Yes: **ALI ALOUI

---

## [Author Response · Author response to Decision Letter 0]

6 Mar 2024

Dear Dr. Elsahoryi, 

I would like to submit our revised manuscript (submission ID: PONE-D-23-42649) entitled “Arabic Translation and Cultural Adaptation of a Training Load And Player Monitoring in High-Level Football Questionnaire: A Cognitive Interview Study” for review and consideration for publication in PLOS ONE. 

I would like to thank the editor and the reviewers for providing constructive feedback on my manuscript. I have addressed each comment individually below. I uploaded a copy of the manuscript with the changes tracked in addition to the manuscript file with all changes accepted. I believe these changes have strengthened the paper, and I look forward to the next steps in the review process.

This research was supported by the Deanship of Scientific Research, Vice Presidency for Graduate Studies and Scientific Research, King Faisal University, Saudi Arabia [Project No. GRANT1445]. The funders had no role in study design, data collection and analysis, decision to publish, or preparation of the manuscript.

Sincerely,

Abdulmalek Bursais

---

## [Editor Report · Decision Letter 1]

26 Mar 2024

Arabic Translation and Cultural Adaptation of a Training Load And Player Monitoring in High-Level Football Questionnaire: A Cognitive Interview Study

PONE-D-23-42649R1

Dear Dr. Bursais,

We’re pleased to inform you that your manuscript has been judged scientifically suitable for publication and will be formally accepted for publication once it meets all outstanding technical requirements.

Kind regards,

Nour Amin Elsahoryi, pHD

Academic Editor

PLOS ONE
---

## [Editor Report · Acceptance letter]

5 Apr 2024

PONE-D-23-42649R1 

PLOS ONE

Dear Dr. Bursais, 

I'm pleased to inform you that your manuscript has been deemed suitable for publication in PLOS ONE. Congratulations! Your manuscript is now being handed over to our production team.

Kind regards, 

on behalf of

Dr. Nour Amin Elsahoryi 

Academic Editor

PLOS ONE